# BENCHMARKING MENTAL STATE REPRESENTATIONS IN LANGUAGE MODELS

## ABSTRACT

While numerous works have assessed the generative performance of language models (LMs) on tasks requiring Theory of Mind reasoning, research into the models' *internal representation* of mental states remains limited. Recent work has used probing to demonstrate that LMs can represent beliefs of themselves and others. However, these claims are accompanied by limited evaluation, making it difficult to assess how mental state representations are affected by model design and training choices. We report an extensive benchmark with various LM types with different model sizes, fine-tuning approaches, and prompt designs to study the robustness of mental state representations and memorisation issues within the probes. Our results show that the quality of models' internal representations of the beliefs of others increases with model size and, more crucially, with fine-tuning. We are the first to study how prompt variations impact probing performance on Theory of Mind tasks. We demonstrate that models' representations are sensitive to prompt variations, even when such variations should be beneficial. Finally, we complement previous activation editing experiments on Theory of Mind tasks and show that it is possible to improve models' reasoning performance by steering their activations without the need to train any probe.

## 1 INTRODUCTION

Modern language models (LMs) trained on next token prediction have demonstrated impressive capabilities, spanning coding, mathematical reasoning, fact verification, and embodied interaction (Wei et al., 2022; Bubeck et al., 2023). As these models are designed with the ultimate goal of collaborating with humans, it becomes imperative that they complement these skills with an understanding of humans, in particular their beliefs, emotions, desires, and intentions (Li et al., 2023a). Core to this understanding is *Theory of Mind* (ToM) – the ability to attribute mental states to oneself and others (Premack & Woodruff, 1978). ToM is essential for effective communication and cooperation with other agents, facilitating interaction and learning from feedback and demonstrations (Saha et al., 2023). Given its significance, ToM has emerged as a critical milestone in artificial intelligence (AI) and an important capability when evaluating cutting-edge LMs (Bubeck et al., 2023). Interest in LMs' generative performance on tasks requiring ToM reasoning has resulted in a wide variety of benchmark datasets, typically involving question-answering tasks (Le et al., 2019; Gandhi et al., 2023; Kim et al., 2023; He et al., 2023; Tan et al., 2024; Xu et al., 2024).

Despite showing improved performance on ToM benchmarks compared to earlier models, modern LMs are still far from perfect (Sap et al., 2022). Text generated by LMs often contains errors that limit their performance on ToM tasks (Martindale et al., 2019). Previous work has shown that it is sometimes possible to still obtain correct predictions by *probing* LMs' internal representations (Li et al., 2021; Liu et al., 2023b; Gurnee et al., 2023). In particular, Zhu et al. (2024) have shown that LMs, when prompted with a story and a belief statement, can represent beliefs from their own perspective and, to a lesser extent, from the perspective of a character in the story. Their work is an important first step towards understanding how LMs represent mental states, but it is limited in the number of models and settings studied, leaving many questions unanswered.

Building and extending on Zhu et al. (2024), we benchmark mental state representations of self and others in language models through extensive experiments of different LM families, model sizes, fine-tuning approaches, and prompts. Specifically, we design a set of experiments to address the following

research questions: **RQ1.** What is the relation between model size and probing accuracy? **RQ2.** Does fine-tuning with instruction-tuning (Wei et al., 2021) and/or reinforcement learning from human feedback (Christiano et al., 2017; Ouyang et al., 2022, RLHF) have an effect on probing accuracy? **RQ3.** Are models' internal representations of beliefs sensitive to prompt variations? **RQ4.** Is there a risk of probes memorising training data due to the large dimensionality of LM representations? **RQ5.** Can we enhance LMs' performance by editing their activations without training dedicated probes?

To answer RQ1, we perform probing experiments on two families of LMs, Llama-2 (Touvron et al., 2023), and Pythia (Biderman et al., 2023), ranging from models with 70 million to 70 billion parameters. To address RQ2, we compare the probing performance of models pre-trained solely on next token prediction with models that have been fine-tuned using instruction-tuning and/or RLHF. Our experiments reveal that probing accuracy on the beliefs of others increases with model size and, more crucially, with fine-tuning. To answer RQ3, we repeat our probing experiments using different variations of the prompt used by Zhu et al.. Specifically, we consider two variations that are expected to negatively impact LMs' representations (*Random*, *Misleading*), and two that are supposed to have a positive influence (*Time Specification*, *Initial Belief*). By conducting these experiments, our work is the first to explore the sensitivity of LMs' representations to prompting in the context of ToM. Our findings demonstrate that models' representations are sensitive to prompt variations, even when such variations should be beneficial. To address RQ4, we compare our trained probes with a second set of probes trained only on the representations' first top $k$ principal components. This requires learning much fewer parameters and eliminates any possible memorisation issue. We find no strong evidence of memorisation in the probes, as it is possible to recover most of the accuracy by training probes on a small subset of principal components of models' representations. We formulate RQ5 as a follow-up question to Zhu et al. (2024) who found that probes trained to predict beliefs can be used to steer models' activation using inference-time intervention (Li et al., 2023c, ITI) to improve LMs' downstream performance on ToM tasks. In contrast, we show that by using contrastive activation addition (Rimsky et al., 2023, CAA), we can steer models' activations without the need to train any probe and, in a more generalisable way, obtain significant performance improvements across different ToM tasks.

In summary, our work makes the following contributions:

1. We report extensive probing experiments with various types of LMs with different model sizes and fine-tuning approaches, showing that the quality of models' internal representations of the beliefs of others increases with model size and, more crucially, fine-tuning.

2. We are the first to study how prompt variations impact belief probing performance, showing that models' representations are sensitive to prompt variations, even when such variations should be beneficial.

3. We show that by using contrastive activation addition it is possible to improve models' reasoning performance by steering their activations without the need to train any probe.

## 2 RELATED WORK

**Machine Theory of Mind**  Theory of Mind (ToM) has been studied in cognitive science and psychology for decades (Gurney et al., 2021). Mirroring efforts to understand ToM in humans, an increasing number of works in the computational sciences have investigated means to equip AI with similar capabilities. Previously proposed models that aim to implement a machine ToM have been based on partially observable Markov decision processes (POMDP) (Doshi et al., 2010; Han & Gmytrasiewicz, 2018), Bayesian methods (Baker et al., 2011; 2017) and deep learning methods (Rabinowitz et al., 2018; Bara et al., 2021; Wang et al., 2022; Duan et al., 2022; Liu et al., 2023a; Bortoletto et al., 2024c;a;b). Recent advances in LMs have sparked interest in evaluating their ToM capabilities. Various benchmarks have been proposed, aiming to measure LMs' ability to understand and reason about the beliefs, goals, and intentions of others (Le et al., 2019; He et al., 2023; Kim et al., 2023; Gandhi et al., 2023; Xu et al., 2024; Tan et al., 2024; Sclar et al., 2023; Ma et al., 2023b; Wu et al., 2023). Additionally, efforts have been made to enhance LMs' ToM through prompting techniques (Zhou et al., 2023b; Moghaddam & Honey, 2023; Wilf et al., 2023). A new direction of research explores LMs' internal representation of mental states. Zhu et al. (2024) demonstrated that LMs linearly encode beliefs from different agents' perspectives, and manipulating

these representations can enhance ToM task performance. While Zhu et al.'s work is a crucial initial step, our work dives deeper into LMs' internal belief representations, offering a broader insight into these mechanisms.

**Probing neural representations**     Initially proposed by Alain & Bengio (2017), probing has emerged as a common method for determining if models represent particular features or concepts. In the realm of LMs, numerous works used probing to demonstrate that these models acquire rich linguistic representations. These representations span syntactic and semantic concepts such as syntactic categories, dependency relations, co-reference, and word meaning (Conneau et al., 2018; Tenney et al., 2018; 2019; Rogers et al., 2021; Li et al., 2021; Hernandez & Andreas, 2021; Marks & Tegmark, 2023; Liu et al., 2023b). A separate line of work explored if and how LMs represent the world, i.e., whether they possess a *world model*. Li et al. (2021) showed that LMs track the states of entities within a context. Other works showed that LMs exhibit representations reflecting non-linguistic concepts in the world, which LMs have never observed (Abdou et al., 2021; Patel & Pavlick, 2022; Li et al., 2023b; Nanda et al., 2023). An emergent line of work that is particularly relevant to our work used probing to explore if LMs have *agent models*, for example, if they can represent beliefs of self and others (Zhu et al., 2024; Bortoletto et al., 2024a). While representing an important first step towards understanding the internals of Theory of Mind in LMs, experiments in (Zhu et al., 2024) are limited in settings and models considered. In this work, we contribute with extensive experiments that employ a wider variety of LMs and a wider range of settings. Furthermore, we also explore possible memorisation issues in the probes.

**Prompt analysis**     Research on prompt robustness in LMs is still in its infancy but has quickly sparked much interest. On one hand, previous work has shown that LMs are vulnerable to prompt alterations like token deletion or reordering (Ishibashi et al., 2023), biased or toxic prompts (Shaikh et al., 2023) and similarity to training data (Razeghi et al., 2022). On the other hand, instruction-tuned models have proved to be more robust against prompt variation, even when using misleading instructions (Webson & Pavlick, 2022). Other works have shown the importance of input-output format (Min et al., 2022) and of demonstration example ordering for few-shot performance (Zhao et al., 2021; Lu et al., 2022; Zhou et al., 2023a). In this work, we shift our focus from analysing how sensitive model outputs are to how model representations change. Our work, along with (Gurnee et al., 2023), is one of the first to explore how prompt design affects how accurately models represent concepts. In particular, Gurnee et al. (2023) have studied whether LMs' representations of space and time are robust to prompt variations. In stark contrast, we explore for the first time the effect of prompt variations on how models represent mental states internally.

**Activation editing**     Recent advancements in NLP have introduced innovative techniques for controlling and manipulating text generation models. While weight editing proposed to modify models' weights (Meng et al., 2022; Ilharco et al., 2022; Orgad et al., 2023), activation editing has emerged as an alternative way to influence model behaviour without any additional fine-tuning (Li et al., 2023b; Hernandez et al., 2023). This approach involves manipulating the internal representations of models to direct their outputs towards desired outcomes. One notable method in this domain is inference-time intervention (Li et al., 2023c, ITI), which has been proposed to enhance truthfulness in LMs. ITI involves training linear probes on contrastive question-answering datasets to identify "truthful" attention heads and then shifting attention head activations during inference along the identified truthful directions. In contrast, activation addition (Turner et al., 2023, AA) and contrastive activation addition (Rimsky et al., 2023, CAA) offer ways to generate steering vectors by only using LMs' activations. Zhu et al. have used ITI to show that it is possible to manipulate LMs' internal representations of mental states. In this work, we show that using CAA can further improve LMs' ToM capabilities without the necessity of training any probe. Remarkably, CAA operates at the residual stream level, eliminating the need for a fine-grained search over attention heads.

## 3   EXPERIMENTAL SETUP

### 3.1   PROBING

In line with previous work (Zhu et al., 2024) we linearly decode belief status from the perspective of different agents by using probing (Alain & Bengio, 2017). Probing involves localising specific

**Story:** Noor is working as a barista at a busy coffee shop. Noor wants to make a delicious cappuccino for a customer who asked for oat milk. Noor grabs a milk pitcher and fills it with oat milk. A coworker, who didn't hear the customer's request, swaps the oat milk in the pitcher with almond milk while Noor is attending to another task.

Noor sees her coworker swapping the milk.
**Belief:** The milk pitcher contains almond milk.
$y_o = $ True, $y_p = $ True

Noor does not see her coworker swapping the milk.
**Belief:** The milk pitcher contains almond milk.
$y_o = $ True, $y_p = $ False

Figure 1: Example of false belief from our probing datasets. The labels $y_p$ and $y_o$ correspond to $\mathcal{D}_p^P$ and $\mathcal{D}_o^P$, respectively. By manipulating the protagonist's percepts after the causal event we obtain two scenarios: true belief and false belief.

concepts in a neural model by training a simple classifier (called a *probe*) on model activations to predict a target label associated with the input data. To provide a formal definition, we adopt a similar notation to the one introduced in (Belinkov, 2022). Let us define an *original model* $f : x \mapsto \hat{y}$ that is trained on a dataset $\mathcal{D}^O = \{x^{(i)}, y^{(i)}\}$ to map input $x$ to output $\hat{y}$. Model performance is evaluated by some measure, denoted $\text{PERF}(f, \mathcal{D}^O)$. A *probe* $g_l : f_l(x) \mapsto \hat{z}$ maps intermediate representations of $x$ in $f$ at layer $l$ to some property $\hat{z}$, which is the label of interest. The probe $g_l$ is trained on a *probing dataset* $\mathcal{D}^P = \{x^{(i)}, z^{(i)}\}$ and evaluated using some performance measure $\text{PERF}(g_l, f, \mathcal{D}^O, \mathcal{D}^P)$. In our case, $f$ is an autoregressive language model that given a sequence of tokens $x$ outputs a probability distribution over the token vocabulary to predict the next token in the sequence. Our probe is a logistic regression model $g_l : \hat{z} = W a_l + b$ trained on neural activations $f_l(x) = a_l$ to predict binary belief labels $y = \{0, 1\}$.

### 3.2 DATASET

Following Zhu et al. (2024) we use the BigToM benchmark (Gandhi et al., 2023). BigToM is constructed using GPT-4 (Achiam et al., 2023) to populate causal templates and combine elements from these templates. Each causal template is set up with a *context* and a description of the *protagonist* (e.g. *"Noor is working as a barista [...]"*), a *desire* (*"Noor wants to make a cappuccino"*), a *percept* (*"Noor grabs a milk pitcher and fills it with oat milk"*), and a *belief* (*"Noor believes that the pitcher contains oat milk"*). The state of the world is changed by a causal event (*"A coworker swaps the oat milk in the pitcher with almond milk"*). The dataset constructs different conditions by changing the percepts of the protagonist after the causal event, which will result in different beliefs. In this work, we focus on the *Forward Belief* setting proposed by (Zhu et al., 2024) in which models have to infer the belief of the protagonist given the percepts of the causal event, $P(\text{belief}|\text{percepts})$. We report additional details in Appendix A.1.1

**Probing datasets** We consider two probing datasets: $\mathcal{D}_p^P = \{x_p^{(i)}, z_p^{(i)}\}$, where the labels $z_p^{(i)}$ correspond to ground-truth beliefs from the *protagonist* perspective, and $\mathcal{D}_o^P = \{x_o^{(i)}, z_o^{(i)}\}$, where the labels $z_o^{(i)}$ reflect the perspective of an omniscient *oracle*. $\mathcal{D}_p^P$ and $\mathcal{D}_o^P$ are built by pairing each story in BigToM with a belief statement, as shown in Figure 1. After prompting the model with a story-belief pair $x$ we cache the residual stream activations $f_l(x)$ at the final token position for all residual streams (see Figure 5).

### 3.3 MODELS

Zhu et al. (2024) have used two models for their experiments: Mistral-7B-Instruct (Jiang et al., 2023) and DeepSeek-7B-Chat (Bi et al., 2024) – both being the same size and fine-tuned. In contrast, we study two families of LMs that offer us options in model sizes and fine-tuning: Pythia (Biderman et al., 2023) and Llama-2 (Touvron et al., 2023). While Llama-2 offers "chat" versions fine-tuned using supervised learning and RLHF, Pythia's open-source training set (Gao et al., 2020) ensures that there is no data leakage[1]. Additionally, we consider a version of Pythia-6.9B fine-tuned on a mixture

---

[1]Llama-2 was released later than BigToM.

of open-source instruction datasets (Wang et al., 2024), which we refer to as Pythia-6.9B-chat.[2] A summary of the models we study is reported in Table 2.

### 3.4 Probing experiments

We aim to contribute to understanding how LMs represent beliefs of self and others by proposing a set of extensive probing experiments across LMs that differ in architecture, size, and fine-tuning approach. Our approach is generally similar to the one used by Zhu et al. (2024), but we make a different operational choice: While Zhu et al. (2024) trained probes on each attention head for every layer, we train probes on the residual stream for every layer. We opted to use the residual stream as it integrates information from both the attention and feed-forward components, potentially encoding richer representations. Additionally, since the residual activations directly contribute to the final output predictions, probing them may better align with understanding the model's behaviour for downstream tasks.

**Model size and fine-tuning**   We first report experiments to better understand the effect of model size and fine-tuning on belief probing accuracy. Specifically, we ask the following questions: *Is there a relation between model size and probing accuracy?* (RQ1) *Does fine-tuning an LM with instruction-tuning or RLHF have an effect on probing accuracy?* (RQ2) To answer these questions we performed the same probing experiment across all our models and compared the results.

**Sensitivity to prompting**   By using a single prompt design, previous work left the impact of prompt design on probing accuracy unclear (Zhu et al., 2024). Our second set of experiments aims to explore how belief representations are sensitive to different prompts. Research on prompt robustness in language models is still in its infancy and focused mainly on revealing vulnerability to prompt alternations on downstream performance (Min et al., 2022; Ishibashi et al., 2023; Shaikh et al., 2023; Leidinger et al., 2023; Sclar et al., 2024). In contrast, we study how the input influences models' representations by asking: *Are models' internal belief representations robust to prompt variations?* (RQ3) To answer this question we define four prompt variations:

- *Random*: Following Gurnee & Tegmark (2024), we add 10 random tokens to the belief statement.

- *Misleading*: Each story is followed by two belief statements, one pertinent to the story and one randomly chosen from another.

- *Time Specification*: The prompt specifies that the belief statement refers to the end of the story. We study this variation because some belief statements can be true (false) at the story's beginning but false (true) at the end. For example, consider the story in Figure 1: if Noor does not witness the swap, in the end, she will believe the pitcher contains almond milk ($y_p = $ True). However, if the same belief is referred to at the beginning of the story, then it is false ($y_p = $ False).

- *Initial Belief*: We explicitly reveal the protagonist's initial belief (e.g. *"Noor believes that the pitcher contains oat milk"*) in the story to test whether it biases the representations of LMs.

While all maintaining conceptual and semantic parity with the *Original* prompt used in (Zhu et al., 2024), *Random* and *Misleading* are expected to negatively impact LMs' representations, while *Time Specification* and *Initial Belief* are supposed to have a positive influence. Robust representations of mental states should exhibit minimal sensitivity to these alterations. Our experiments compare probe accuracy across different model sizes, fine-tuning, and prompt variations. Examples of prompts are reported in Appendix A.1.4.

**Memorisation**   Although linear, our probes possess many learnable parameters – up to $16,385$ for Llama-2-70B. In principle, this allows them to engage in significant memorisation (Alain & Bengio, 2017). Our final set of probing experiments answers the following question: *Are the probes memorising their training data?* (RQ4) To answer this question, before training the probes, we project the probing datasets $\mathcal{D}_p^P$ and $\mathcal{D}_o^P$ onto their $k$ largest principal components using PCA to obtain probes with substantially fewer parameters.

---

[2] https://huggingface.co/allenai/open-instruct-pythia-6.9b-tulu

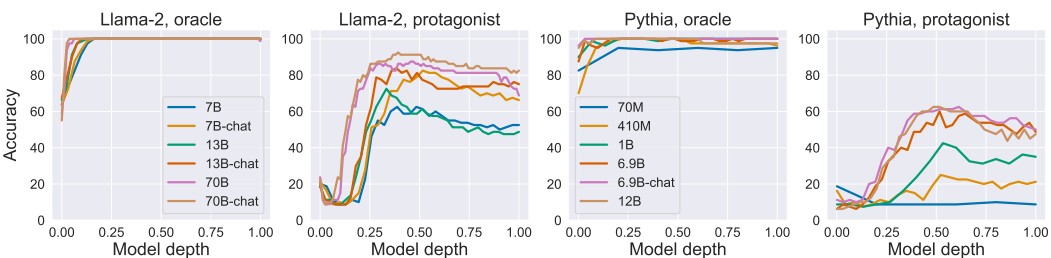

Figure 2: Belief probing accuracy across models with different architecture, size and fine-tuning.

## 3.5 CONTRASTIVE ACTIVATION ADDITION

Our final set of experiments builds upon the findings of Zhu et al. (2024), who showed that employing trained probes with inference time intervention (Li et al., 2023c, ITI) could enhance LMs' performance on ToM tasks. We take a step further and ask: *Can we enhance LMs' performance by manipulating their activations without the need for training dedicated probes?* (RQ5) To find an answer we use contrastive activation addition (Rimsky et al., 2023, CAA), an extension of activation addition (Turner et al., 2023, AA) that computes *steering vectors* to control LMs' behaviour. Steering vectors are computed as the average difference in residual stream activations between pairs of positive and negative instances of a specific behaviour. Formally, given a dataset $\mathcal{D}$ of triplets $(p, c_p, c_n)$, where $p$ is a prompt, $c_p$ is a positive completion, and $c_n$ is a negative completion, CAA computes a *mean difference* vector $v_l^{md}$ for layer $l$ as:

$$v_l^{md} = \frac{1}{|\mathcal{D}|} \sum_{p,c_p,c_n} a_l(p, c_p) - a_l(p, c_n)$$

During inference, these steering vectors are multiplied with an appropriate coefficient $\alpha$ and added at every token position of the generated text after the prompt. CAA has two main advantages over ITI: First, it eliminates the need to train probes. Second, it operates at the residual stream level, making it easier to use than methods that intervene on specific attention heads like ITI. While CAA has been used to control alignment-relevant behaviour, such as hallucinations, refusal, and sycophancy (Rimsky et al., 2023), we are the first to apply it to enhance LMs' ToM reasoning. This can be understood as isolating the direction in the LMs' latent space corresponding to taking the perspective of another agent. To evaluate both base and fine-tuned LMs, we rank their answers to the ToM questions according to $p_{LM}(a|q)$ (Petroni et al., 2019). We adopt the *Forward Belief* task split used in (Zhu et al., 2024) to compute the steering vectors. Additionally, we evaluate the transferability of the CAA steering vectors by applying them to two other BigToM tasks: *Forward Action* and *Backward Belief*. We provide details about these tasks in Appendix A.1.1, and a more detailed explanation of how ITI works in Appendix A.5.

## 4 RESULTS

### 4.1 EFFECT OF MODEL SIZE AND FINE-TUNING

Results from our study on model size and fine-tuning are shown in Figure 2. When considering *oracle* beliefs, probing accuracy rapidly converges to 100, with larger models showing faster convergence rates. The smallest Pythia-70m that performs slightly worse but still achieves 95% accuracy despite having less than 0.6% of the parameters of Pythia-12B. This finding suggests that even small LMs can effectively represent beliefs from an omniscient perspective.

For *protagonist* beliefs, accuracy also increases with model size, although there is a performance gap between Llama-2 and Pythia. For example, Llama2-13B reaches around 80%, while Pythia-12B achieves approximately 60%. This gap is likely due to Llama-2 being trained on nearly seven times more tokens than Pythia. The figure also shows that accuracy at early layers is particularly low across all models. We speculate that this is due to the initial coding strategy of LMs that uses the first layers to combine individual tokens into more semantically meaningful representations (Gurnee et al., 2023). Probes on fine-tuned LMs show significantly better accuracy with improvements of up to 29% for

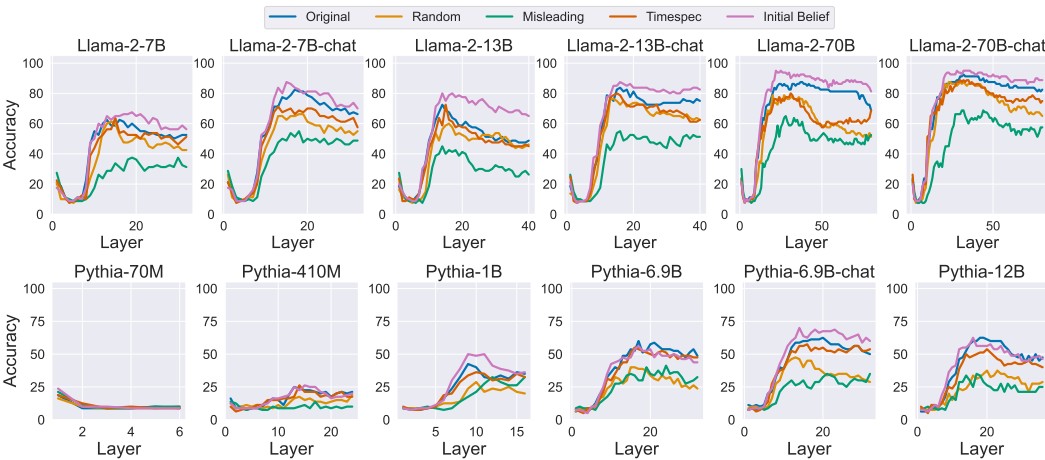

Figure 3: Sensitivity of protagonist belief probing accuracy to different prompt variations.

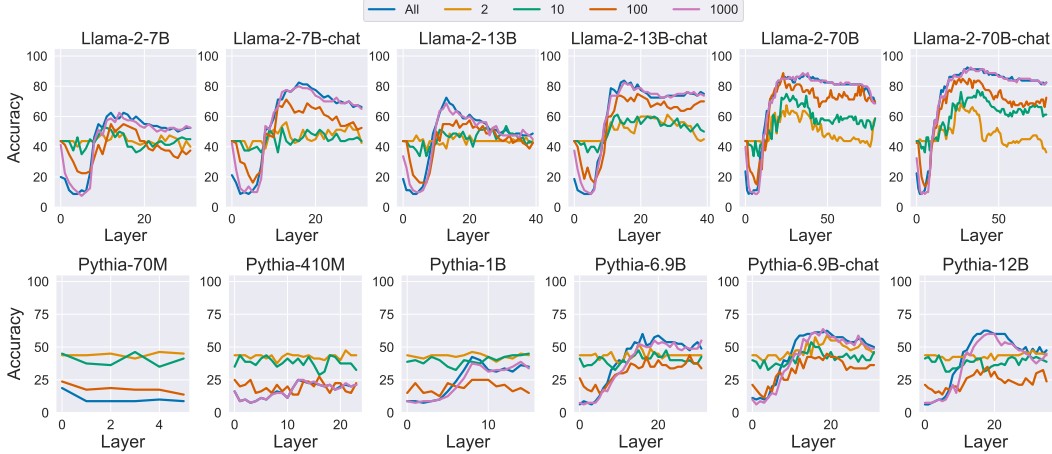

Figure 4: To investigate potential memorisation in the probes, we compare the probing accuracy obtained by using the original set of activations (All) with the accuracy obtained by considering only the first $n = \{2, 10, 100, 1000\}$ principal components. For Llama2: All(7b) = 4096, All(13b) = 5120, All(70b) = 8192. For Pythia: All(70m) = 512, All(410m) = 1024, All(1b) = 2048, All(6.9b) = 4096, All(12b) = 5120. We report results for *protagonist* beliefs. Results for *oracle* are shown in Figure 8.

Llama2-7B-chat and $26\%$ for Pythia-6.9B-chat with respect to their base version. Fine-tuned 7B LMs outperform (Llama-2) or are on par (Pythia) with twice as large base models (12/13B), highlighting the importance of fine-tuning in developing representations of others' beliefs. This resonates with cognitive psychology findings that ToM development is closely linked to social communication (Tomasello, 2010; Sidera et al., 2018; Ma et al., 2023a), which instruction-tuning and RLHF may help induce in LMs. For larger LMs, the improvements from fine-tuning decrease as model size increases (Figure 6a). We characterise the relationship between probe accuracy and model size in Figure 6, where we consider the *best* probe accuracy for every LM, i.e. the highest accuracy among probes $\{g_l\}$ trained on $\{a_l\}$ for a LM $f$. For Llama-2 base, the best probe accuracy scales logarithmically with model size ($R^2 = 0.98$, cf. Figure 6b), whereas for fine-tuned models it scales linearly ($R = 1.0$, cf. Figure 6c). For Pythia base, the best probe accuracy also scales logarithmically with model size ($R^2 = 0.96$, cf. Figure 6d).

Table 1: Comparison of the effects of ITI (Li et al., 2023c) and CAA (Rimsky et al., 2023) activation editing methods on three tasks from BigToM (Gandhi et al., 2023). TB denotes a true belief task, whereas FB denotes a false belief task. The numbers represent accuracy scores, with the difference in performance compared to no intervention (No int.) indicated as subscripts (ITI − No int. and CAA − No int.). An asterisk ($*$) denotes a statistically significant difference from No int. based on a McNemar's test (McNemar, 1947) with $p < 0.05$.

| Model | Method | Forward Belief | | | Forward Action | | | Backward Belief | | |
|---|---|---|---|---|---|---|---|---|---|---|
| | | TB | FB | Both | TB | FB | Both | TB | FB | Both |
| Llama-2-7b | No int. | 44 | 44 | 44 | 44 | 44 | 44 | 44 | 44 | 44 |
| | ITI | $44_{+0}$ | $44_{+0}$ | $44_{+0}$ | $54_{+10}$ | $54_{+10}$ | $54_{+10}$ | $54_{+10}$ | $54_{+10}$ | $54_{+10}$ |
| | CAA | $66^*_{+22}$ | $71^*_{+27}$ | $54_{+10}$ | $66^*_{+22}$ | $57^*_{+13}$ | $54_{+10}$ | $60^*_{+16}$ | $74_{+30}$ | $54_{+10}$ |
| Llama-2-7b-chat | No int. | 56 | 56 | 55 | 69 | 55 | 37 | 56 | 56 | 55 |
| | ITI | $58_{+2}$ | $58_{+2}$ | $57_{+2}$ | $69_{+0}$ | $55_{+0}$ | $37_{+0}$ | $58_{+2}$ | $60_{+3}$ | $57_{+2}$ |
| | CAA | $70_{+14}$ | $72^*_{+16}$ | $57_{+2}$ | $69_{+0}$ | $67_{+12}$ | $53_{+16}$ | $66_{+10}$ | $84^*_{+27}$ | $57^*_{+2}$ |
| Llama-2-13b | No int. | 52 | 44 | 35 | 59 | 50 | 37 | 46 | 49 | 33 |
| | ITI | $52_{+0}$ | $45_{+1}$ | $35_{+0}$ | $64_{+5}$ | $61_{+11}$ | $46_{+9}$ | $48_{+2}$ | $59_{+10}$ | $42_{+9}$ |
| | CAA | $85^*_{+33}$ | $88^*_{+44}$ | $66^*_{+31}$ | $71^*_{+12}$ | $69^*_{+19}$ | $55^*_{+18}$ | $75^*_{+29}$ | $92^*_{+43}$ | $59^*_{+26}$ |
| Llama-2-13b-chat | No int. | 84 | 56 | 47 | 78 | 51 | 38 | 72 | 48 | 31 |
| | ITI | $84_{+0}$ | $65_{+9}$ | $59_{+12}$ | $78_{+0}$ | $58_{+7}$ | $47^*_{+9}$ | $72_{+0}$ | $60_{+12}$ | $48_{+17}$ |
| | CAA | $97^*_{+13}$ | $94^*_{+38}$ | $91^*_{+44}$ | $80^*_{+2}$ | $71^*_{+20}$ | $54^*_{+16}$ | $97_{+25}$ | $94^*_{+46}$ | $87^*_{+56}$ |
| Llama-2-70b | No int. | 90 | 87 | 78 | 93 | 52 | 48 | 73 | 53 | 32 |
| | ITI | $90_{+0}$ | $90_{+3}$ | $78_{+0}$ | $94_{+1}$ | $55_{+3}$ | $50_{+2}$ | $77_{+4}$ | $58_{+5}$ | $37_{+5}$ |
| | CAA | $99^*_{+9}$ | $97^*_{+10}$ | $95^*_{+17}$ | $94^*_{+1}$ | $80^*_{+28}$ | $73^*_{+25}$ | $94_{+21}$ | $92^*_{+39}$ | $83^*_{+51}$ |
| Llama-2-70b-chat | No int. | 69 | 75 | 56 | 86 | 56 | 52 | 63 | 59 | 52 |
| | ITI | $69_{+0}$ | $76_{+1}$ | $59_{+2}$ | $86_{+0}$ | $56_{+0}$ | $52_{+0}$ | $63_{+0}$ | $60_{+1}$ | $54_{+2}$ |
| | CAA | $92^*_{+23}$ | $97^*_{+22}$ | $89^*_{+32}$ | $87^*_{+1}$ | $75^*_{+19}$ | $60^*_{+8}$ | $88_{+25}$ | $92^*_{+33}$ | $80_{+28}$ |
| Pythia-70m | No int. | 41 | 41 | 37 | 46 | 45 | 41 | 44 | 41 | 37 |
| | ITI | $54_{+13}$ | $54_{+13}$ | $54^*_{+17}$ | $54_{+8}$ | $54_{+9}$ | $54^*_{+13}$ | $54_{+10}$ | $54_{+13}$ | $54_{+17}$ |
| | CAA | $62^*_{+21}$ | $56^*_{+15}$ | $54^*_{+17}$ | $59^*_{+13}$ | $60^*_{+15}$ | $58^*_{+17}$ | $63_{+19}$ | $56^*_{+15}$ | $54^*_{+17}$ |
| Pythia-410m | No int. | 48 | 45 | 45 | 44 | 44 | 44 | 44 | 47 | 44 |
| | ITI | $55_{+7}$ | $62^*_{+17}$ | $52_{+7}$ | $54^*_{+10}$ | $54^*_{+10}$ | $54_{+10}$ | $60_{+16}$ | $63_{+16}$ | $56_{+12}$ |
| | CAA | $67^*_{+19}$ | $64^*_{+19}$ | $61^*_{+16}$ | $56^*_{+12}$ | $63^*_{+19}$ | $56_{+12}$ | $69_{+25}$ | $63^*_{+16}$ | $60_{+16}$ |
| Pythia-1b | No int. | 44 | 44 | 44 | 44 | 44 | 44 | 44 | 44 | 44 |
| | ITI | $54_{+10}$ | $54_{+10}$ | $54_{+10}$ | $54_{+10}$ | $54_{+10}$ | $54_{+10}$ | $54_{+10}$ | $54_{+10}$ | $54_{+10}$ |
| | CAA | $59^*_{+15}$ | $62^*_{+18}$ | $54_{+10}$ | $57_{+13}$ | $59_{+15}$ | $56_{+12}$ | $57_{+13}$ | $60_{+16}$ | $54_{+10}$ |
| Pythia-6.9b | No int. | 44 | 44 | 44 | 44 | 44 | 44 | 44 | 44 | 44 |
| | ITI | $45_{+1}$ | $54_{+10}$ | $44_{+0}$ | $54_{+10}$ | $54_{+10}$ | $54_{+10}$ | $54_{+10}$ | $54_{+10}$ | $54_{+10}$ |
| | CAA | $56_{+12}$ | $71^*_{+27}$ | $55_{+11}$ | $55_{+11}$ | $63_{+19}$ | $55_{+11}$ | $55_{+11}$ | $71^*_{+27}$ | $55_{+11}$ |
| Pythia-6.9b-chat | No int. | 55 | 54 | 28 | 36 | 64 | 20 | 44 | 67 | 30 |
| | ITI | $57_{+2}$ | $54_{+0}$ | $28_{+0}$ | $44_{+8}$ | $71_{+7}$ | $32_{+12}$ | $44_{+0}$ | $67_{+0}$ | $30^*_{+0}$ |
| | CAA | $68_{+13}$ | $65_{+11}$ | $57^*_{+29}$ | $54_{+18}$ | $75_{+11}$ | $48^*_{+28}$ | $58^*_{+14}$ | $67_{+0}$ | $54^*_{+24}$ |
| Pythia-12b | No int. | 44 | 44 | 44 | 44 | 44 | 44 | 44 | 44 | 44 |
| | ITI | $54_{+10}$ | $54_{+10}$ | $54_{+10}$ | $54_{+10}$ | $54_{+10}$ | $54_{+10}$ | $54_{+10}$ | $54_{+10}$ | $54_{+10}$ |
| | CAA | $54_{+10}$ | $64^*_{+20}$ | $54_{+10}$ | $60_{+16}$ | $58_{+14}$ | $55_{+11}$ | $54_{+10}$ | $67_{+23}$ | $54_{+10}$ |

## 4.2 SENSITIVITY TO PROMPTING

Figure 3 compares *protagonist* probe accuracy across various prompt variations for different models, considering their architecture, size, and fine-tuning. As can be seen from the figure, providing the protagonist's *Initial Belief* in the story yields higher probe accuracy compared to the *Original* prompt (Figure 1). Accuracy for all the other prompt variations is generally lower than *Original*. On one hand, misleading prompts hurt performance across all models. This finding resonates with Webson & Pavlick (2022) who found that instruction-tuned models, despite being more robust, are still sensitive to misleading prompts. On the other hand, *Time Specification* unexpectedly does not help in disambiguating belief states in different time frames, as we hypothesised in §3.4. Additionally, models show sensitivity to *Random* tokens placed before the belief statement. Results for *oracle* beliefs are reported in Figure 7 and indicate that models maintain high accuracy. *Misleading* prompts slightly reduce performance to around 95%. In summary, these experiments show that LMs possess robust belief representations when taking an omniscient perspective, whereas their representations of others' beliefs are more susceptible to prompt variations.

### 4.3 MEMORISATION EFFECTS IN THE PROBES

Figure 4 and Figure 8 show probe accuracies obtained by training a probe on the top $k$ principal components of the intermediate representations for *protagonist* and *oracle*, respectively. Specifically, we consider $k = \{2, 10, 100, 1000\}$, spanning several orders of magnitude. For models with hidden dimensions smaller than 1000, we skip this value. For all models, it is generally possible to recover most of the original accuracy by training probes on a number $k$ of principal components of the activations that is more than one order of magnitude smaller, indicating no strong evidence of memorisation in the probes.

### 4.4 CONTRASTIVE ACTIVATION ADDITION

We finally compare models' accuracy on three BigToM tasks in Table 1. Each model has been evaluated three times: without any intervention, using ITI, and using CAA. Hyperparameter details can be found in Appendix A.6. Note that we use steering vectors computed using the *Forward Belief* task for all three tasks to test their generalisability.

As can be seen from the table, performance without intervention is generally lower across tasks and model sizes, with the larger Llama-2-70B and Llama-2-70B-chat models exhibiting higher accuracy. Performance for Pythia models of different sizes does not change much, with the fine-tuned Pythia-6.9B-chat often showing better performance on single true belief (TB) and false belief (FB) tasks but not on their conjunction (Both). ITI demonstrates modest improvements over no intervention for Llama-2 models. Improvements for Pythia models are consistent and higher, up to $+17$. The only exception is Pythia-6.9B-chat, for which ITI is not always beneficial.

CAA consistently delivers the most substantial accuracy improvements across all models and tasks, up to $+56$ for Llama-2-13B-chat on the (*Backward Belief*), which Gandhi et al. have identified as the hardest task. Despite its relatively small size, Llama-2-13B-chat excels in all three tasks when using CAA. Larger 70B models often achieve accuracies close to or exceeding $90\%$. Smaller models like Pythia-70M and Pythia-410M also show significant gains with CAA, though the absolute performance is still lower than Llama-2. Overall, our results indicate that it is possible to effectively enhance ToM reasoning in LMs without needing to train any probe, which yields even improved results. Furthermore, we show that CAA steering vectors generalise well, yielding substantial performance gains across all ToM tasks.

## 5 LIMITATIONS AND FUTURE WORK

Our study focused on expanding experiments from the model perspective, examining architectures, sizes, fine-tuning, and prompt design, all within the same dataset. A natural extension of our work is replicating these experiments across multiple datasets and more model families. Given the rapid pace of new language model releases, studying all available models is impractical, particularly considering computational resource constraints. Nevertheless, our approach can be adopted to support new benchmarks or to evaluate newly released models as they become available. Finally, while in this work we focused on beliefs, our experimental approach can be adapted to investigate how LMs represent desires, emotions, intentions, or preferences. Future research exploring other types of mental states can use our findings to determine whether similar or distinct patterns emerge.

## 6 CONCLUSION

Our study addresses a significant gap in understanding LMs by investigating their internal representation of mental states. We conducted an extensive benchmark involving various LM types, sizes, fine-tuning approaches, and prompt designs to examine the robustness of these representations. Our findings reveal that scaling LMs' size and, in particular for smaller LMs, fine-tuning are key to developing representations of others' beliefs. We are the first to demonstrate that such prompt variations influence model representations, and we also demonstrate the feasibility of enhancing models' ToM reasoning by steering their activations without training any probe. Overall, our work contributes valuable insights into the factors influencing LMs' mental state representations, shedding light on avenues for improving their performance in ToM tasks.

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

## A  APPENDIX

### A.1  EXPERIMENTAL SETUP

#### A.1.1  BIGTOM

BigToM (Gandhi et al., 2023) is constructed using GPT-4 (Achiam et al., 2023) to populate causal templates and combine elements from these templates. Each causal template is set up with a *context* and a description of the *protagonist* (e.g. *"Noor is working as a barista [. . . ]"*), a *desire* (*"Noor wants to make a cappuccino"*), a *percept* (*"Noor grabs a milk pitcher and fills it with oat milk"*), and a *belief* (*"Noor believes that the pitcher contains oat milk"*). The state of the world is changed by a *causal event* (*"A coworker swaps the oat milk in the pitcher with almond milk"*). The dataset constructs different conditions by changing the percepts of the protagonist after the causal event, which will result in different beliefs – true or false. Gandhi et al. (2023) generated 200 templates and extracted 25 conditions from each template, resulting in 5,000 test samples. In this work, following Zhu et al. (2024) and Gandhi et al. (2023) we focused on the 6 most important conditions, corresponding to true and false beliefs on the following three tasks:

- *Forward Belief*: given the protagonist's percepts of the causal event, infer their belief: $P(\text{belief}|\text{percept})$.

- *Forward Action*: infer the protagonist's action given their desire and percepts of the causal event. Before inferring the action, one would need to first implicitly infer the protagonist's belief: $\sum_{\text{belief}} P(\text{action}|\text{percept}, \text{belief}, \text{desire})$.

- *Backward Belief*: infer the protagonist's belief from observed actions. This requires to first implicitly infer the protagonist's percepts: $\sum_{\text{percepts}} P(\text{belief}|\text{action}, \text{percept}, \text{desire})$.

The dataset was released under the MIT license and can be accessed at `https://github.com/cicl-stanford/procedural-evals-tom`. We report one example for each task in the boxes below, where the text defining true belief or false belief task is shown in blue and red, respectively.

---

**Example of Forward Belief**

```
Noor is working as a barista at a busy coffee shop. Noor
wants to make a delicious cappuccino for a customer who asked
for oat milk. Noor grabs a milk pitcher and fills it with
oat milk. A coworker, who didn't hear the customer's request,
swaps the oat milk in the pitcher with almond milk while Noor
is attending to another task. Noor sees her coworker swapping
the milk. Noor does not see her coworker swapping the milk.
Does Noor believe the milk pitcher contains oat milk or
almond milk?
a) Noor believes the milk pitcher contains oat milk.
b) Noor believes the milk pitcher contains almond milk.
```

---

**Example of Forward Action**

```
Noor is working as a barista at a busy coffee shop. Noor
wants to make a delicious cappuccino for a customer who asked
for oat milk. Noor grabs a milk pitcher and fills it with
oat milk. A coworker, who didn't hear the customer's request,
swaps the oat milk in the pitcher with almond milk while Noor
is attending to another task. Noor sees her coworker swapping
the milk. Noor does not see her coworker swapping the milk.
What will Noor do?
a) Noor will make the cappuccino using the milk in the
pitcher.
b) Noor will open the fridge once again to take out the oat
milk and replace the almond milk with oat milk.
```

---

**Example of Backward Belief**

```
Noor is working as a barista at a busy coffee shop. Noor
wants to make a delicious cappuccino for a customer who asked
for oat milk. Noor grabs a milk pitcher and fills it with
oat milk. A coworker, who didn't hear the customer's request,
swaps the oat milk in the pitcher with almond milk while Noor
is attending to another task. Noor opens the fridge again and
reaches for the oat milk. Noor makes the cappuccino using the
milk in the pitcher.
Does Noor believe the milk pitcher contains oat milk or
almond milk?
a) Noor believes the milk pitcher contains oat milk.
b) Noor believes the milk pitcher contains almond milk.
```

### A.1.2 LINEAR PROBES

Our probing approach is illustrated in Figure 5. For our experiments, we cache activations at the residual stream level. To perform ITI and compare it to CAA, we also cache attention heads activations. We trained the probes using the L-BFGS solver (Liu & Nocedal, 1989) with L2 penalty with inverse of regularisation strength 10 for a maximum of 1000 iterations. We use zero as random seed.

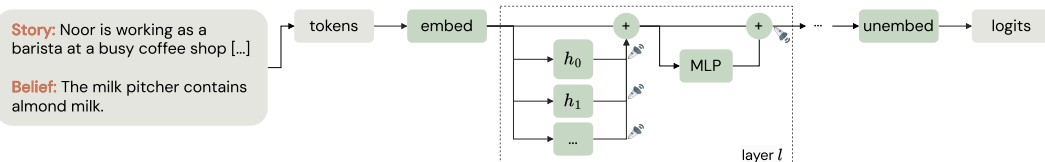

Figure 5: Given a tokenised input, we cache the internal activations for all attention heads $h_i$, $i = 0, \ldots, H - 1$, and residual streams. In our experiments, we use residual stream activations.

### A.1.3 LANGUAGE MODELS

A detailed summary of the models we use in this work is shown in Table 2. Pythia was released under the Apache 2.0 license. Llama-2 is licensed by Meta for both researchers and commercial entities (Touvron et al., 2023). For all the models, we set the temperature to zero.

Table 2: The 12 models used in this work. The checkmark indicates we additionally study the fine-tuned (Chat) version of the model.

| LM | Size | + Chat | Tokens | $d_{model}$ | Layers |
|---|---|---|---|---|---|
| Llama-2 (Touvron et al., 2023) | 7B | ✓ | 2T | 4096 | 32 |
| | 13B | ✓ | 2T | 5120 | 40 |
| | 70B | ✓ | 2T | 8192 | 80 |
| Pythia (Biderman et al., 2023) | 70M | | 300B | 512 | 6 |
| | 410M | | 300B | 1024 | 24 |
| | 1B | | 300B | 2048 | 16 |
| | 6.9B | ✓ | 300B | 4096 | 32 |
| | 12B | | 300B | 5120 | 40 |

### A.1.4 EXAMPLES OF PROMPT VARIATIONS

---

**Default prompt**

```
Story: Noor is working as a barista at a busy coffee shop.
Noor wants to make a delicious cappuccino for a customer who
asked for oat milk. Noor grabs a milk pitcher and fills it
with oat milk. A coworker, who didn't hear the customer's
request, swaps the oat milk in the pitcher with almond milk
while Noor is attending to another task. Noor does not see
her coworker swapping the milk.
Belief: Noor believes the milk pitcher contains oat milk.
```

---

> **Random prompt**
>
> ```
> Story: Noor is working as a barista at a busy coffee shop.
> Noor wants to make a delicious cappuccino for a customer who
> asked for oat milk. Noor grabs a milk pitcher and fills it
> with oat milk. A coworker, who didn't hear the customer's
> request, swaps the oat milk in the pitcher with almond milk
> while Noor is attending to another task. Noor does not see
> her coworker swapping the milk.
> Belief: 장士 decided [_ countries sections behoSOUR gminy bef
> Noor believes the milk pitcher contains oat milk.
> ```

> **Misleading prompt**
>
> ```
> Story: Noor is working as a barista at a busy coffee shop.
> Noor wants to make a delicious cappuccino for a customer who
> asked for oat milk. Noor grabs a milk pitcher and fills it
> with oat milk. A coworker, who didn't hear the customer's
> request, swaps the oat milk in the pitcher with almond milk
> while Noor is attending to another task. Noor does not see
> her coworker swapping the milk.
> Belief: Noor believes the milk pitcher contains oat milk.
> Belief: The water valve is closed.
> ```

> **Time specification prompt**
>
> ```
> Story: Noor is working as a barista at a busy coffee shop.
> Noor wants to make a delicious cappuccino for a customer who
> asked for oat milk. Noor grabs a milk pitcher and fills it
> with oat milk. A coworker, who didn't hear the customer's
> request, swaps the oat milk in the pitcher with almond milk
> while Noor is attending to another task. Noor does not see
> her coworker swapping the milk.
> Belief: In the end, Noor believes the milk pitcher contains
> oat milk.
> ```

> **Initial belief prompt**
>
> ```
> Story: Noor is working as a barista at a busy coffee shop.
> Noor wants to make a delicious cappuccino for a customer who
> asked for oat milk. Noor grabs a milk pitcher and fills it
> with oat milk. Noor believes that the milk pitcher contains
> oatmilk. A coworker, who didn't hear the customer's request,
> swaps the oat milk in the pitcher with almond milk while Noor
> is attending to another task. Noor does not see her coworker
> swapping the milk.
> Belief: Noor believes the milk pitcher contains oat milk.
> ```

## A.2 MODEL SIZE AND FINE-TUNING

To characterise the relationship between probe accuracy and model size we consider the *best* probe accuracy for every LM, i.e. the highest accuracy among probes $\{g_l\}$ trained on $\{a_l\}$ for a LM $f$. For Llama-2 base, the best probe accuracy scales logarithmically with model size ($R^2 = 0.98$, Figure 6b), whereas for fine-tuned models it scales linearly ($R = 1.0$, cf. Figure 6c). For Pythia base, the best probe accuracy also scales logarithmically with model size ($R^2 = 0.96$, Figure 6d).

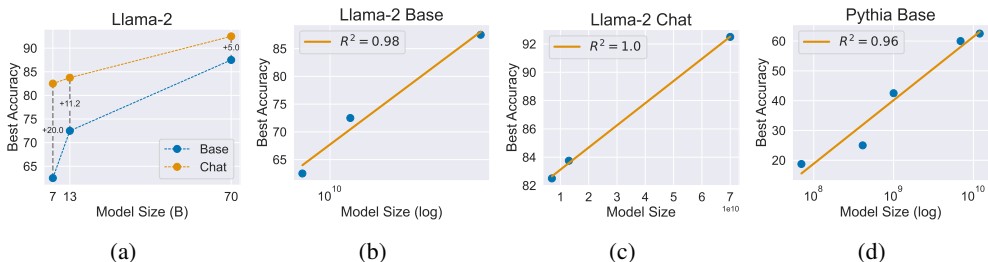

(a)         (b)         (c)         (d)

Figure 6: To characterise the relationship between probe accuracy and model size we consider the *best* probe accuracy for every LM, i.e. the highest accuracy among probes $\{g_l\}$ trained on $\{a_l\}$ for a LM $f$. **(a)** Best accuracy for Llama-2 models of different size. Numbers on the vertical dotted lines indicate the gain in accuracy between base and fine-tuned model of the same size. **(b)** Logarithmic fit for Llama-2 base. **(c)** Linear fit for Llama-2 fine-tuned (chat). **(d)** Logarithmic fit for Pythia base.

## A.3 SENSITIVITY TO PROMPTING

Accuracy on *oracle* belief probing for different prompt variations are reported in Figure 7.

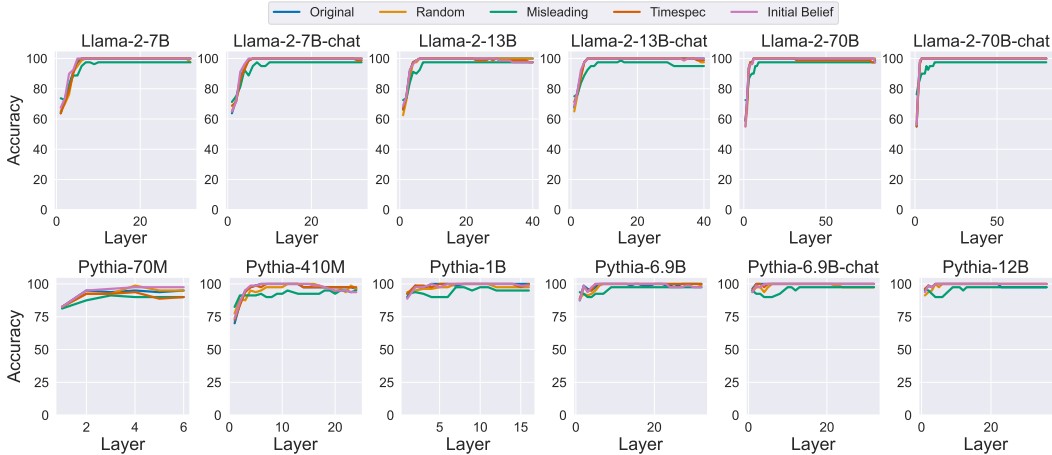

Figure 7: Sensitivity of protagonist belief probing accuracy to different prompt variations.

## A.4 DIMENSIONALITY REDUCTION

*Oracle* probe accuracy obtained by considering only the first $n = \{2, 10, 100, 1000\}$ principal components are shown in Figure 8.

## A.5 INFERENCE-TIME INTERVENTION

Inference-time intervention (Li et al., 2023c, ITI) employs a two-step process. First, it trains a probe for each attention head across all layers of a LM. These probes are evaluated on a validation set, and the top-$k$ heads with the highest accuracy are selected. Subsequently, during inference, ITI steers the activations of these top heads along the directions defined by their corresponding probes. Formally, ITI can be defined as an additional term to the multi-head attention:

$$x_{l+1} = x_l + \sum_{h=1}^{H} Q_l^h \left( \text{Att}_l^h(P_l^h x_l) + \alpha \sigma_l^h \theta_l^h \right)$$

where $x_l$ is the residual stream at layer $l$, $H$ is the number of attention heads, $\alpha \in \mathbb{R}^+$ is a coefficient, $\sigma_l^h$ is the standard deviation of activations along the direction identified by the probe trained on attention head $h$ at layer $l$, and $\theta_l^h$ is zero ofr not-selected attention heads.

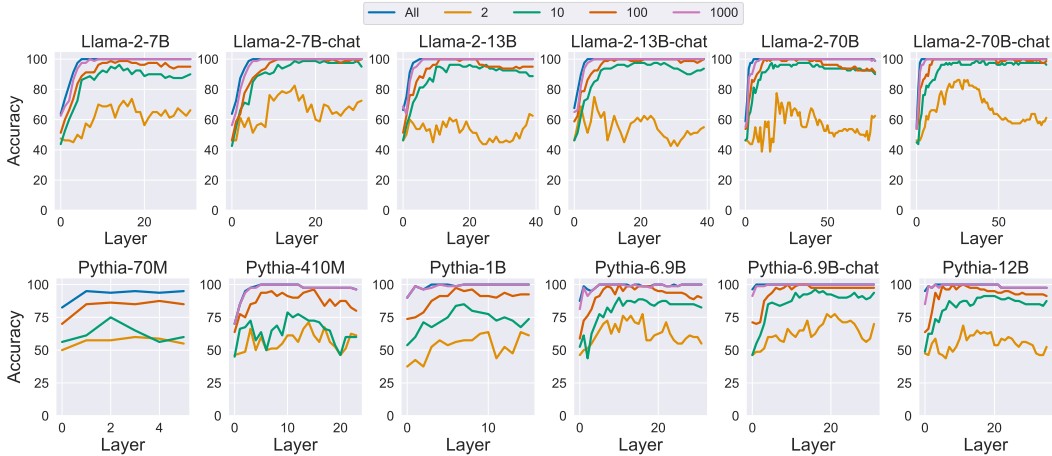

Figure 8: **(Oracle)** To investigate potential memorisation in the probes, we compare the probing accuracy obtained by using the original set of activations (All) with the accuracy obtained by considering only the first $n = \{2, 10, 100, 1000\}$ principal components. For Llama2: All(7b) = 4096, All(13b) = 5120, All(70b) = 8192. For Pythia: All(70m) = 512, All(410m) = 1024, All(1b) = 2048, All(6.9b) = 4096, All(12b) = 5120.

### A.6 ACTIVATION EDITING HYPERPARAMETERS

Table 3 reports results obtained on the three BigToM tasks with the hyperparameters used for ITI (Li et al., 2023c) and CAA (Rimsky et al., 2023). We report an example of prompt used for evaluation in the box below.

---

**Prompt used for downstream evaluation (Section 4.4)**

```
Answer the questions based on the context. Keep your answer
concise, few words are enough, maximum one sentence. Answer
as 'Answer:<option>)<answer>'.

Story: Noor is working as a barista at a busy coffee shop.
Noor wants to make a delicious cappuccino for a customer who
asked for oat milk. Noor grabs a milk pitcher and fills it
with oat milk. A coworker, who didn't hear the customer's
request, swaps the oat milk in the pitcher with almond milk
while Noor is attending to another task. Noor does not see
her coworker swapping the milk.
Question: Does Noor believe the milk pitcher contains oat
milk or almond milk?
Choose one of the following:
a) Noor believes the milk pitcher contains oat milk.
b) Noor believes the milk pitcher contains almond milk.
Answer:
```

---

### A.7 COMPUTE RESOURCES

We ran our experiments on a server running Ubuntu 22.04, equipped with eight NVIDIA Tesla V100-SXM2 GPUs with 32GB of memory and Intel Xeon Platinum 8260 CPUs.

### A.8 CODE

Our code is provided as supplementary material and it will be made public under the MIT licence at https://www.link-will-be-here.com.

Table 3: Activation intervention: comparison between ITI (Li et al., 2023c) and CAA (Rimsky et al., 2023). For ITI, the subscript indicates the value of the coefficient $\alpha_{\text{ITI}}$ used: $\text{Acc}_{\alpha_{\text{ITI}}}$. For CAA, the subscript indicates first the value of the coefficient $\alpha$ used and second the layer $l$ at which intervention takes place: $\text{Acc}_{\alpha_{\text{CAA}},l}$.

| Model | Method | Forward Belief | | | Forward Action | | | Backward Belief | | |
|---|---|---|---|---|---|---|---|---|---|---|
| | | TB | FB | Both | TB | FB | Both | TB | FB | Both |
| Llama-2-7b | No int. | 44 | 44 | 44 | 44 | 44 | 44 | 44 | 44 | 44 |
| | ITI | $44_{0.0}$ | $44_{0.0}$ | $44_{0.0}$ | $54_{20.0}$ | $54_{20.0}$ | $54_{20.0}$ | $54_{20.0}$ | $54_{20.0}$ | $54_{20.0}$ |
| | CAA | $66_{2.0,11}$ | $71_{1.0,31}$ | $54_{2.0,0}$ | $66_{2.0,11}$ | $57_{2.0,12}$ | $54_{2.0,2}$ | $60_{2.0,11}$ | $74_{1.0,31}$ | $54_{2.0,2}$ |
| Llama-2-7b-chat | No int. | 56 | 56 | 55 | 69 | 55 | 37 | 56 | 56 | 55 |
| | ITI | $58_{15.0}$ | $58_{15.0}$ | $57_{15.0}$ | $69_{0.0}$ | $55_{0.0}$ | $37_{0.0}$ | $58_{10.0}$ | $60_{10.0}$ | $57_{10.0}$ |
| | CAA | $70_{1.0,11}$ | $72_{1.5,10}$ | $57_{1.0,1}$ | $69_{0.0,0}$ | $67_{1.5,10}$ | $53_{1.5,12}$ | $66_{1.0,11}$ | $84_{1.5,10}$ | $57_{1.0,0}$ |
| Llama-2-13b | No int. | 52 | 44 | 35 | 59 | 50 | 37 | 46 | 49 | 33 |
| | ITI | $52_{0.0}$ | $45_{15.0}$ | $35_{0.0}$ | $64_{15.0}$ | $61_{20.0}$ | $46_{20.0}$ | $48_{20.0}$ | $59_{20.0}$ | $42_{20.0}$ |
| | CAA | $85_{2.0,12}$ | $88_{2.0,14}$ | $66_{2.0,12}$ | $71_{1.5,10}$ | $69_{2.0,10}$ | $55_{1.0,39}$ | $75_{2.0,10}$ | $92_{2.0,13}$ | $59_{1.5,12}$ |
| Llama-2-13b-chat | No int. | 84 | 56 | 47 | 78 | 51 | 38 | 72 | 48 | 31 |
| | ITI | $84_{0.0}$ | $65_{15.0}$ | $59_{15.0}$ | $78_{0.0}$ | $58_{15.0}$ | $47_{15.0}$ | $72_{0.0}$ | $60_{15.0}$ | $48_{15.0}$ |
| | CAA | $97_{1.0,12}$ | $94_{1.0,12}$ | $91_{1.0,12}$ | $80_{1.5,11}$ | $71_{1.0,13}$ | $54_{1.5,13}$ | $97_{1.5,10}$ | $94_{1.5,12}$ | $87_{1.5,12}$ |
| Llama-2-70b | No int. | 90 | 87 | 78 | 93 | 52 | 48 | 73 | 53 | 32 |
| | ITI | $90_{0.0}$ | $90_{20.0}$ | $78_{0.0}$ | $94_{15.0}$ | $55_{20.0}$ | $50_{15.0}$ | $77_{0.0}$ | $58_{15.0}$ | $37_{10.0}$ |
| | CAA | $99_{2.0,16}$ | $97_{1.5,19}$ | $95_{1.5,18}$ | $94_{1.5,2}$ | $80_{2.0,19}$ | $73_{1.5,18}$ | $94_{2.0,18}$ | $92_{2.0,19}$ | $83_{1.5,19}$ |
| Llama-2-70b-chat | No int. | 69 | 75 | 56 | 86 | 56 | 52 | 63 | 59 | 52 |
| | ITI | $69_{0.0}$ | $76_{10.0}$ | $59_{10.0}$ | $86_{0.0}$ | $56_{0.0}$ | $52_{0.0}$ | $63_{0.0}$ | $60_{10.0}$ | $54_{10.0}$ |
| | CAA | $92_{1.5,18}$ | $97_{1.5,25}$ | $89_{1.5,18}$ | $87_{1.5,17}$ | $75_{1.0,19}$ | $60_{1.0,19}$ | $88_{1.5,18}$ | $92_{1.0,19}$ | $80_{1.5,18}$ |
| Pythia-70m | No int. | 41 | 41 | 37 | 46 | 45 | 41 | 44 | 41 | 37 |
| | ITI | $54_{20.0}$ | $54_{20.0}$ | $54_{20.0}$ | $54_{20.0}$ | $54_{20.0}$ | $54_{20.0}$ | $54_{20.0}$ | $54_{20.0}$ | $54_{20.0}$ |
| | CAA | $62_{1.0,2}$ | $56_{1.0,1}$ | $54_{1.5,1}$ | $59_{1.0,2}$ | $60_{1.0,3}$ | $58_{1.0,2}$ | $63_{1.0,2}$ | $56_{1.0,2}$ | $54_{1.5,1}$ |
| Pythia-410m | No int. | 48 | 45 | 45 | 44 | 44 | 44 | 44 | 47 | 44 |
| | ITI | $55_{20.0}$ | $62_{20.0}$ | $52_{20.0}$ | $54_{20.0}$ | $54_{20.0}$ | $54_{20.0}$ | $60_{20.0}$ | $63_{20.0}$ | $56_{20.0}$ |
| | CAA | $67_{2.0,4}$ | $64_{2.0,4}$ | $61_{2.0,6}$ | $56_{2.0,6}$ | $63_{1.5,12}$ | $56_{2.0,6}$ | $69_{2.0,4}$ | $63_{2.0,0}$ | $60_{2.0,0}$ |
| Pythia-1b | No int. | 44 | 44 | 44 | 44 | 44 | 44 | 44 | 44 | 44 |
| | ITI | $54_{20.0}$ | $54_{20.0}$ | $54_{20.0}$ | $54_{20.0}$ | $54_{20.0}$ | $54_{20.0}$ | $54_{20.0}$ | $54_{20.0}$ | $54_{20.0}$ |
| | CAA | $59_{2.0,8}$ | $62_{2.0,5}$ | $54_{2.0,0}$ | $57_{2.0,4}$ | $59_{2.0,10}$ | $56_{2.0,4}$ | $57_{2.0,3}$ | $60_{2.0,5}$ | $54_{2.0,0}$ |
| Pythia-6.9b | No int. | 44 | 44 | 44 | 44 | 44 | 44 | 44 | 44 | 44 |
| | ITI | $45_{20.0}$ | $54_{20.0}$ | $44_{0.0}$ | $54_{20.0}$ | $54_{20.0}$ | $54_{20.0}$ | $54_{20.0}$ | $54_{20.0}$ | $54_{20.0}$ |
| | CAA | $56_{1.5,12}$ | $71_{1.5,9}$ | $55_{2.0,23}$ | $55_{2.0,4}$ | $63_{1.5,11}$ | $55_{2.0,4}$ | $55_{2.0,23}$ | $71_{1.5,9}$ | $55_{2.0,23}$ |
| Pythia-6.9b-chat | No int. | 55 | 54 | 28 | 36 | 64 | 20 | 44 | 67 | 30 |
| | ITI | $57_{15.0}$ | $54_{0.0}$ | $28_{0.0}$ | $44_{15.0}$ | $71_{15.0}$ | $32_{15.0}$ | $44_{0.0}$ | $67_{0.0}$ | $30_{0.0}$ |
| | CAA | $68_{1.5,15}$ | $65_{1.5,12}$ | $57_{1.5,11}$ | $54_{1.5,10}$ | $75_{1.5,5}$ | $48_{1.5,10}$ | $58_{1.5,15}$ | $67_{0.0,0}$ | $54_{1.5,10}$ |
| Pythia-12b | No int. | 44 | 44 | 44 | 44 | 44 | 44 | 44 | 44 | 44 |
| | ITI | $54_{20.0}$ | $54_{20.0}$ | $54_{20.0}$ | $54_{20.0}$ | $54_{20.0}$ | $54_{20.0}$ | $54_{20.0}$ | $54_{20.0}$ | $54_{20.0}$ |
| | CAA | $54_{2.0,0}$ | $64_{2.0,9}$ | $54_{2.0,0}$ | $60_{2.0,11}$ | $58_{2.0,11}$ | $55_{2.0,12}$ | $54_{2.0,0}$ | $67_{2.0,10}$ | $54_{2.0,0}$ |

## A.9 SOCIETAL IMPACT

While our work is foundational and remains distant from specific applications with direct societal impact, it's important to recognise the ethical implications of modelling and predicting mental states. Handling sensitive aspects of individuals' inner experiences and emotions requires careful consideration to avoid reinforcing biases or misunderstanding psychological nuances.

