# OpenReview forum: "Benchmarking Mental State Representations in Language Models"
_ICLR.cc/2025/Conference — ICLR 2025 Conference Withdrawn Submission_

### Official Review · Reviewer_Jtvy · 2024-10-29

**Soundness:** 3
**Presentation:** 3
**Contribution:** 3
**Rating:** 5
**Confidence:** 4

**Summary:**

This paper investigates how language models (LMs) internally represent mental states, such as beliefs, in Theory of Mind (ToM) tasks. The study benchmarks multiple LMs across varying model sizes, fine-tuning methods, and prompt designs to explore the robustness of mental state representations. It addresses several research questions, including the relationship between model size and probing accuracy, the impact of fine-tuning on performance, the sensitivity of internal representations to prompt variations, and the potential risks of memorization within probes.

**Strengths:**

* This paper investigates how the internal representations of mental states and beliefs within language models evolve with changes in model size and training stages, providing valuable insights into the factors that influence these representations.
* The study examines how prompt variations affect the probing performance of LMs on Theory of Mind tasks, offering important findings on the sensitivity of models to different prompt designs.

**Weaknesses:**

1. On RQ2: The exploration of "fine-tuning with instruction-tuning and/or RLHF") is not sufficiently convincing. Different datasets and strategies for fine-tuning can lead to varying changes in the models' mental state representations. However, the paper only compares the base and chat versions of open-source models. I encourage the authors to provide a more detailed analysis in this area.

2. On Line 091 the statement "It is possible to improve models’ reasoning performance by steering their activations" may cause misunderstanding. The term reasoning here specifically refers to Theory of Mind reasoning and not to performance on tasks such as GSM8K or other general reasoning benchmarks.

**Questions:**

In Figure 7, the accuracy of the protagonist belief detection appears to be insensitive to variations in different prompts. You mentioned in line 431 that "LMs possess robust belief representations when taking an omniscient perspective." Could you provide more references to analyze the possible reasons for this phenomenon?

---

> ### Author Response · Authors · 2024-11-20
>
> We thank the reviewer for the feedback. We are pleased to hear that they find the insights we offer valuable. We address their concerns and questions below.
>
> > The paper only compares the base and chat versions of open-source models. I encourage the authors to provide a more detailed analysis in this area.
>
> We use open-source models because unfortunately it is not possible to access internal activations of closed-source models.
>
> > Line 091 the statement "It is possible to improve models’ reasoning performance by steering their activations" may cause misunderstanding.
>
> Thank you, we will change it to “It is possible to improve models’ performance on theory of mind tasks by steering their activations”.
>
> > In Figure 7, the accuracy of the protagonist's belief detection appears to be insensitive to variations in different prompts. You mentioned in line 431 that "LMs possess robust belief representations when taking an omniscient perspective." Could you provide more references to analyze the possible reasons for this phenomenon?
>
> We apologise for the typo in the caption of Figure 7. It should be “Sensitivity of oracle belief probing accuracy”. We believe that LMs possess robust belief representations when taking an omniscient perspective because that makes the task easier. In particular, such a scenario is much easier because the LLM does not have to take the protagonist’s perspective.
> \
> \
> We kindly ask if the reviewer could consider whether our clarifications support an increase in their score.

---

> > ### Comment · Reviewer_Jtvy · 2024-11-23
> >
> > For weakness 1, your research question focuses on "fine-tuning with instruction-tuning and/or RLHF." However, you have not explored key comparative aspects, such as unaligned models and well-aligned models. Instead, you only utilize broad categories like the base model and chat model, which may oversimplify the analysis.

---

> ### Author Response · Authors · 2024-11-23
>
> We thank the reviewer for the response. Could the reviewer please clarify what they mean by unaligned/well-aligned?

---

> > ### Comment · Reviewer_Jtvy · 2024-11-23
> >
> > For "fine-tuning with instruction-tuning and/or RLHF," I believe the models should include: a base model that has not undergone instruction-tuning (base model), a model that has undergone instruction-tuning but lacks alignment with human values (unaligned chat model), and a model that has been aligned with human values (aligned chat model). Therefore, I am puzzled by the choice of only selecting the base model and the chat model in your experiments, because the human value alignment stage will highly influence the  Mental State Representations of LLM.

---

> > > ### Author Response · Authors · 2024-11-23
> > >
> > > Thank you for the clarification. Unfortunately the "unaligned chat model" is not available for Llama 2, and instruction-tuning the base models ourselves is not possible (the datasets are not publicly available). However, the finetuned version of Pythia-6.9B is an "unaligned chat model" (L214). Therefore, we selected all the 3 variants the reviewer is asking for, but unfortunately having the 3 variants for the same model family was not possible. We will make this distinction clearer in our revised document.
> > >
> > > In terms of findings, our results show that probes trained on the SFT version of Pythia-6.9B perform on par with probes trained on the much bigger 12B base model (see Figure 2, fourth plot and L368). These results are analogous to what we found when comparing Llama 2 models finetuned with SFT+RLHF with base models (Figure 2, second plot and L368).

---

### Official Review · Reviewer_cSmu · 2024-11-04

**Soundness:** 2
**Presentation:** 2
**Contribution:** 2
**Rating:** 3
**Confidence:** 4

**Summary:**

This paper explores the Theory of Mind performance of LLMs across various settings, specifically focusing on their internal representations to address five research questions related to model size, tuning, prompts, memorization, and inference-time intervention. Through empirical experiments, it provides valuable observations and comparative results.

**Strengths:**

- This work provides useful information on different LLM settings and their implications for Theory of Mind performance.
- This work conducts extensive experiments to reveal diverse behavioral aspects of LLMs in relation to Theory of Mind.

**Weaknesses:**

- While these empirical results provide useful information, they do not appear to offer significant insights.
- The experiments addressing the research questions seem more like incremental extensions of previous work.
- The research questions are scattered rather than interconnected, making it difficult to grasp the main ideas of the paper.
- It would be more beneficial if the paper focused on one or two research questions and investigated them in greater depth.

**Questions:**

- How did the authors conduct the experiment for the pre-trained base model? Did they use few-shot demonstrations, or did they simply provide instructions, treating it the same as an instruction-tuned model?

---

> ### Author Response · Authors · 2024-11-20
>
> We thank the reviewer for the feedback. We are happy that they appreciate our extensive experiments and find our findings useful. We address their concerns and questions below.
>
> > While these empirical results provide useful information, they do not appear to offer significant insights.
>
> We challenge this claim. Our work represents the first attempt to organise extensive benchmarking experiments to study theory of mind in LLMs from a different perspective than pure downstream generative performance. We do not only perform probing experiments but we interpret them (e.g. L318-322). We find that bigger models in general produce better probing accuracy. We then find that fine-tuned models can represent others’ beliefs with high accuracy even with smaller size, which suggests that instruction-tuning or RLHF – being linked to social communication – is important for training advanced models that can reason about humans to cooperate with them. While this might sound trivial, it is not until someone proves it. We believe that this insight on the relationship between RLHF and representations of mental states will open up new directions for future research. \
> Prompting plays also a crucial part in LLMs’ behaviour: we show that prompts that are supposed to help the model to construct the correct representation of the protagonist’s mental states do not always do it (see Fig. 3, bottom) or yield marginal benefit (see Fig. 3, top). This insight is significant for studying new prompt engineering approaches. Moreover, our results suggest that LLMs’ responses may be heavily influenced by superficial features (e.g. random additional tokens, see yellow line in Fig. 3). This raises broader implications for deploying LLMs in scenarios where reliable mental state modelling is critical, such as education or psychology. \
> Related to this last statement, we found that using a minimally invasive and computationally cheap technique such as CAA we can drastically and consistently improve LLM downstream generative performance and generalisability. This insight is extremely significant, for example, for future research working on deployable LLM applications. \
> We are happy to make these insights even clearer in the camera-ready.
>
> > The experiments addressing the research questions seem more like incremental extensions of previous work.
>
> We do not see building on previous work as a negative. It is, in fact, a cornerstone of scientific progress. While some of our experiments can be seen as extension of previous work, some are not, and most importantly we offer new insights on LLMs’ representations of others’ mental states.
>
> > The research questions are scattered rather than interconnected, making it difficult to grasp the main ideas of the paper.
>
> Our response to the first question clearly outlines the interconnection between our research questions and a grasp of the main ideas of the paper.
>
> > How did the authors conduct the experiment for the pre-trained base model? Did they use few-shot demonstrations, or did they simply provide instructions, treating it the same as an instruction-tuned model?
>
> Base models are not trained to follow instructions, so instead we rank answers according to their log-probability.
> \
> \
> We kindly ask if the reviewer could consider whether our clarifications support an increase in their score.

---

> > ### Comment · Reviewer_cSmu · 2024-11-23
> >
> > Thank you for the response. However, after reviewing the authors' response, I remain unconvinced about the impact of the proposed benchmarking. While it provides useful information, it offers limited insight. Therefore, I will maintain my original score.

---

> > > ### Author Response · Authors · 2024-11-25
> > >
> > > We thank the reviewer for the response. We would like to ask how exactly our paper offers limited insight.
> > >
> > > Our paper offers significant insights by addressing several research gaps in the study of how language models represent mental states of others (L54):
> > >
> > > * *There is no work studying the relation between model size and probing accuracy (RQ1).* Our experiments across two model families (Pythia and Llama 2) and different model sizes (70m to 70B) reveal that probing accuracy increases logarithmically with models' size (L373 + Figure 6).
> > > * *There is no work studying if and how fine-tuning LMs using instruction-tuning and/or RLHF has an effect on probing accuracy (RQ2).* Our findings indicate that probes trained on the representations of fine-tuned LMs achieve significantly higher accuracy. Notably, fine-tuned 7B LMs outperform (Llama-2 with instruction-tuning and RLHF) or match the performance (Pythia with instruction-tuning) of base models with double the parameter count (L368).
> > > * *There is no work studying if and how models’ internal representations of beliefs are sensitive to prompt variations (RQ3).* We conducted our experiments using four different prompt variations and observed that the models’ internal representations lack robustness, showing a decrease in performance even when the prompt helps resolving ambiguity (L427).
> > > * *There is no work comparing ITI with other methods when it comes to steering models' representations of other's mental states (RQ5).* Our results show that it is possible to steer models' activations in a generalisable way by using CAA (section 4.4). CAA delivers substantial improvements across all models and all tasks while at the same time requiring less computational efforts (no training of probes on every single attention head).

---

> ### Comment · Reviewer_cSmu · 2024-11-26
>
> Despite the experimental effort made by the authors, the choices of size, tuning, etc., seem somewhat trivial and do not appear to be substantially related to the core purpose of benchmarking Theory of Mind.

---

> > ### Author Response · Authors · 2024-11-26
> >
> > Thank you for the response. As we write in the abstract and introduction, our core purpose is benchmarking models' *representations*, i.e. assess how mental state representations are affected by model design and training choices (L16). Our research questions and model choices are tailored to this purpose.

---

### Official Review · Reviewer_Qg3X · 2024-11-04

**Soundness:** 3
**Presentation:** 3
**Contribution:** 2
**Rating:** 5
**Confidence:** 4

**Summary:**

This paper studies probing in theory of mind (ToM) to study five research questions, including correlation between model size and probing accuracy, instruction-following fine-tuning for probing accuracy, sensitivity to prompt variations, LLM memorization for probing, and edit activations without training probes. Conducted on the BigToM across Lllama-2 and Pythia model families, results show that probing performance increases with model sizes, especially after fine-tuning. The paper also shows that using contrastive activation addition (CAA), it is possible to improve model performance without training any probe.

**Strengths:**

1. This paper studies five research questions related to probing language models in the context of theory of mind and conducted extensive experiments and drew insightful conclusions. This can inspire future researchers studying related tasks.
2. This paper introduces CAA, which achieves good probing performance without training any probe. This can be interesting to the community exploring probing and interpreting large language models.

**Weaknesses:**

1. This paper raises research questions and introduces methods to probe language models motivated by theory of minds and studies specifically for ToM. However, besides the dataset used, it is not clear to see the connection to ToM. On the positive side, the proposed method can be universally applicable to general probing tasks (however, the research questions and conclusions have been extensively studied, and the methods proposed and experiments conducted do not provide additional observation to the community). On the negative side, it is not convincing that the conclusion in this paper is indeed revealing LLM's mental representations, especially when comparing to previous methods. I would suggest the authors to draw some stronger connection between the proposed research questions and experiments and ToM, and more importantly, how is it different from general probing tasks and previous methods.

**Questions:**

1. Why do you think CAA, by simply computing a mean difference vector, would be sufficient to serve as probing without training? Would this be able to apply to other probing tasks or is it ToM specific? Is there any trend with regard to model size, more model training, etc?

---

> ### Author Response · Authors · 2024-11-20
>
> We thank the reviewer for the feedback. We are happy that they appreciate our extensive experiments and find our findings insightful. We address their concerns and questions below.
>
> > Besides the dataset used, it is not clear to see the connection to ToM.
>
> Any task or evaluation in deep learning inherently depends on the dataset used. The dataset we use defines the scope of the problem being addressed and serves as the foundation for testing models’ abilities to represent mental states.
>
> > It is not convincing that the conclusion in this paper is indeed revealing LLM's mental representations, especially when comparing to previous methods.
>
> We are not sure about which previous methods the reviewer has in mind. We also believe there might be a misunderstanding as the reviewer writes “LLM’s mental representations”, which is not the correct wording. We study to which extent LLMs are capable of  representing the mental states of an individual that is not the LLM itself. We kindly ask the reviewer to reconsider this point.
>
> > Why do you think CAA, by simply computing a mean difference vector, would be sufficient to serve as probing without training? Would this be able to apply to other probing tasks or is it ToM specific? Is there any trend with regard to model size, more model training, etc?
>
> We are afraid there might be another misunderstanding: CAA is not a probing method. CAA is an activation steering method that works for any scenario in which it is possible to construct positive and negative pairs (see L289). \
> Regarding trends, as shown in Table 1 and discussed in Section 4.4, the trend we observe is that CAA delivers substantial improvements across all models and all tasks. In absolute terms, these improvements are similar across all model sizes and fine-tuning. In relative terms, the models with the highest increase are the smallest. We are happy to emphasise this more in the paper.
> \
> \
> We kindly ask if the reviewer could consider whether our clarifications support an increase in their score.

---

> > ### Comment · Reviewer_Qg3X · 2024-11-24
> > **Thanks for the response and clarification.**
> >
> > Regarding mental states in LLMs, sorry for the bad wording. There is no question that "any evaluation inherently depends on the datasets used". However, it is still not clear to me how this reveals how LLMs are capable of representing the mental states, especially how the methods used and conclusions derived in the paper "addresses a significant gap in understanding LMs by investigating their internal representation of mental states."
> >
> > Similarly, please note that the question about how CCA can be used for steering activation (I was not suggesting that it was the same as other probing methods) was a general question regarding how it could be applied to other tasks.

---

> ### Author Response · Authors · 2024-11-24
>
> We thank the reviewer for the response.
>
> > it is still not clear to me how this reveals how LLMs are capable of representing the mental states, especially how the methods used and conclusions derived in the paper "addresses a significant gap in understanding LMs by investigating their internal representation of mental states."
>
> Theory of mind is foundational to human interactions and therefore key in the development advanced AI models capable of cooperating with humans. As we discuss in the paper (L43), LMs have been evaluated on ToM tasks mainly using generative settings, but are still far from perfect and often make mistakes. However, previous work showed that it is still possible to obtain more accurate predictions by probing models' internal activations [1,2,3,4]. We use probing as this is the established method to "inspect" internal representations in neural networks. \
> Previous research has shown that certain language models (LMs) can, to some extent, internally represent the mental states of others [4]. However, the two LMs analyzed in [4] share the same number of parameters (7B) and are both instruction-tuned, leaving unexplored whether these findings generalize to models with different architectures or training paradigms. Additionally, there has been no investigation into how robust these mental state representations are when presented with varied prompts.
> These are the research gaps that our paper addresses (L54):
> * *There is no work studying the relation between model size and probing accuracy (RQ1).* Our experiments across two model families (Pythia and Llama 2) and different model sizes (70m to 70B) reveal that probing accuracy increases logarithmically with models' size (L373 + Figure 6).
> * *There is no work studying if and how fine-tuning LMs using instruction-tuning and/or RLHF has an effect on probing accuracy (RQ2).*  Our findings indicate that probes trained on the representations of fine-tuned LMs achieve significantly higher accuracy. Notably, fine-tuned 7B LMs outperform (Llama-2 with instruction-tuning and RLHF) or match the performance (Pythia with instruction-tuning) of base models with double the parameter count (L368).
> * *There is no work studying if and how models’ internal representations of beliefs are sensitive to prompt variations (RQ3).* We conducted our experiments using four different prompt variations and observed that the models’ internal representations lack robustness, showing a decrease in performance even when the prompt helps resolving ambiguity (L427).
>
> > the question about how CCA can be used for steering activation (I was not suggesting that it was the same as other probing methods) was a general question regarding how it could be applied to other tasks.
>
> As we wrote in our first answer, CAA is an activation steering method that works for any scenario in which it is possible to construct positive and negative pairs (see L289). For example, CAA was originally tested on alignment-relevant tasks, such as
> coordination, corrigibility, hallucination, myopic reward, survival instinct, sycophancy and refusal [5].
>
> [1] Li, Belinda Z., Maxwell Nye, and Jacob Andreas. "Implicit representations of meaning in neural language models." ACL 2021. \
> [2] Liu, Kevin, et al. "Cognitive Dissonance: Why Do Language Model Outputs Disagree with Internal Representations of Truthfulness?." EMNLP 2023. \
> [3] Gurnee, Wes, et al. "Finding neurons in a haystack: Case studies with sparse probing." TMLR 2023. \
> [4] Zhu, Wentao, Zhining Zhang, and Yizhou Wang. "Language Models Represent Beliefs of Self and Others." ICML 2024. \
> [5] Panickssery, Nina, et al. "Steering llama 2 via contrastive activation addition." arXiv preprint arXiv:2312.06681 (2023).

---

### Official Review · Reviewer_pD2v · 2024-11-05

**Soundness:** 2
**Presentation:** 3
**Contribution:** 1
**Rating:** 3
**Confidence:** 4

**Summary:**

This paper is a benchmark for measures of representations of agents' mental states inside LMs from the lens of probing. It studies the effect of model size, finetuning, and prompting variations on probe accuracy, and tests an activation steering method for its ability to improve the model's abilities at relevant tasks.

This work adds some experiments on top of previous probing work on the BigToM dataset.

**Strengths:**

* I think the paper is easy to follow, reasonably well structured, and mostly well written.
* The paper seems to include a pretty comprehensive discussion of related work.

**Weaknesses:**

* The paper is on the whole a bit disjointed—a bunch of results related to theory of mind representations but without clear or enlightening takeaways as far as I can tell. I'm unclear on what is important about these results. Probing accuracy is a very weak measure of representation quality, and we want to see things like steering results to know the results are meaningful. But also: what use might come out of steering even if we do it well? Again it's unclear to me. The activation steering experiments bias the model towards correctly answering the question from the protagonist's view. But what do they do with respect to querying the oracle belief? Does it actually improve model reasoning or just bias it?
* I'm not convinced by the $k$ principal component ablation. I think if you want to show that the probing experiment is meaningful and not just a product of strong high-dimensional features then you probably want something like Hewitt and Liang (2019)'s _control tasks_ (https://aclanthology.org/D19-1275/). But better would be using the probes to do steering. Why not just include steering results for the probes?
* The description of the results at the beginning is confusing. For example: it says the paper demonstrates that the results of probing experiments are sensitive to prompting. This means nothing to me: of course the results will not be exactly the same when the prompts are different; the question is how different and if there are any interesting patterns to this. But the abstract and intro mention the results without saying anything on the matter. The sentence `We demonstrate that models’ representations are sensitive to prompt variations, even when such variations should be beneficial` is confusing—"even when" implies a contrast, but the variations being beneficial would be an example of sensitivity to prompt variation.
* The idea that the probing task on this data measures representations of theory of mind seems very questionable to me. The probe itself is specific to the protagonist of the story—but what if there are multiple protagonists? What if there is none? What's the mechanism by which the probe picks up on the representation of the protagonist's mental state and not some other correlate in the dataset like a feature of the narrative structure in the story (e.g., an indicator of someone seeing an event or missing it)? It seems to me like a weak setting for testing theory of mind. Improving the data would go a long way here I think, whereas the extra experiments in this paper tell us very little in my view.
* The point about steering vectors avoiding the need for training dedicated probes doesn't make sense to me. If you need a training dataset of positive and negative examples to compute a steering vector, you're in the exact same situation as someone training a probe—you're just using a simpler architecture and loss (i.e., your model is just a linear regressor).


Overall I think there are some potential problems with the experimental soundness and major issues with interpretation and impact of the results.

**Questions:**

* Is there a reason you used Llama 2 instead of Llama 3? 3 has been out for a while now.
* I'm confused by RQ4, about whether the probes are memorizing their training data. Does it matter? You evaluate it on held-out test data, right? (If not, that is... bad.) That should tell you all you need to know. If you're worried that the probes are learning the task despite the LM not really knowing it, I agree this is a concern but I think you don't address it. (See my comment about control tasks under Weaknesses.)

---

> ### Author Response · Authors · 2024-11-20
>
> We thank the reviewer for the feedback. We are happy that they found the paper easy to follow, well structured and written, and with a comprehensive discussion of related work. We address their concerns and questions below.
>
> > Importance of the results and use of the steering results
>
> Our work represents the first attempt to organise extensive benchmarking experiments to study theory of mind in LLMs from a different perspective than pure downstream generative performance. We do not only perform probing experiments but we interpret them (e.g. L318-322). We find that bigger models in general produce better probing accuracy. We then find that fine-tuned models can represent others’ beliefs with high accuracy even with smaller size, which suggests that instruction-tuning or RLHF – being linked to social communication – is important for training advanced models that can reason about humans to cooperate with them. While this might sound trivial, it is not until someone proves it. We believe that this insight on the relationship between RLHF and representations of mental states will open up new directions for future research. For example: What specific metrics can be developed to measure the impact of RLHF on ToM performance? Does RLHF improve ToM performance only in collaborative settings or in deceptive settings as well? \
> Prompting also plays a crucial part in LLMs’ behaviour: we show that prompts that are supposed to help the model do not always do it (see Fig. 3, bottom) or yield marginal benefit (see Fig. 3, top). This insight is significant for studying new prompt engineering approaches. Moreover, our results suggest that LLMs’ responses may be heavily influenced by superficial features (e.g. random additional tokens, see yellow line in Fig. 3). This raises broader implications for deploying LLMs in scenarios where reliable mental state modelling is critical, such as education or psychology. \
> Related to this last statement, we found that by using a minimally invasive and computationally cheap technique such as CAA we can drastically and consistently improve LLM downstream generative performance and generalisability. This insight is extremely significant, for example, for future research working on deployable LLM applications. \
> We are happy to make these insights even clearer in the camera-ready.
>
> > Steering activations when querying the oracle belief: Does it actually improve model reasoning or just bias it?
>
> The dataset that we use does not evaluate oracle settings. Given that the task focuses on theory of mind – which implies taking someone else's perspective – oracle settings are less interesting as the task boils down to text comprehension, which current models can already perform well.
>
> > Not convinced by the k principal component ablation. Why not just include steering results for the probes?
>
> Using dimensionality reduction on features is an established method originally proposed by Alain & Bengio [1] and currently in use to this date [2, 4]. \
> We include steering results for inference time intervention with the probes in Table 1 (L378-415), see ITI rows.
>
> > Differences and interesting patterns in different prompts.
>
> Please see our response to the first question.
>
> > The sentence “We demonstrate that models’ representations are sensitive to prompt variations, even when such variations should be beneficial” is confusing.
>
> Thank you, we will rephrase: “We demonstrate that models’ representations are sensitive to prompt variations, with performance decrease or minimal improvements even when such variations are beneficial”.
>
> > The probe itself is specific to the protagonist of the story— multiple protagonists? What if there is none?
>
> Studying multiple protagonists is a very interesting idea for future research, but outside of the scope of our paper. If there is no protagonist, then the task wouldn’t make sense.
>
> > What's the mechanism by which the probe picks up on the representation of the protagonist's mental state and not some other correlate in the dataset like a feature of the narrative structure in the story (e.g., an indicator of someone seeing an event or missing it)?
>
> We agree with the reviewer that, by itself, probing is not a sufficient condition to show that the model is representing the protagonist’s mental state. This is why the probes are then used to perform inference-time intervention. The increase in performance (even if not as high as for CAA) is a strong suggestion of the validity of the probes [4].

---

> > ### Comment · Reviewer_pD2v · 2024-11-30
> > **Thanks for your responses**
> >
> > Sorry for the late response. Let's see if I can reduce the issue to the most important crux. You state (emphasis mine):
> > > We then find that fine-tuned models can represent others’ beliefs with high accuracy even with smaller size, which suggests that instruction-tuning or RLHF – being linked to social communication – is important for training advanced models that can reason about humans to cooperate with them. **While this might sound trivial, it is not until someone proves it.** We believe that this insight on the relationship between RLHF and representations of mental states will open up new directions for future research. For example: What specific metrics can be developed to measure the impact of RLHF on ToM performance? Does RLHF improve ToM performance only in collaborative settings or in deceptive settings as well?
> >
> > I think this is where my core complaint is. I do not think the claim sounds trivial — far from it. And I also don't think the paper has proven it, and that's the problem. What's going on here is a classic problem of construct validity. The paper claims to measure a specific construct: ToM in LMs. In order to justify this claim, the measure needs to be shown to be indicative of this construct as it manifests in a variety of settings — i.e., it needs to be shown that the measure _measures what we think it does_. But tests of ToM in language models are notoriously non-robust, with conflicting and non-robust results across a variety of tests (see [Ullman, 2023](https://arxiv.org/pdf/2302.08399), [Zhou et al., 2023](https://arxiv.org/pdf/2310.03051), [Shapira et al., 2023](https://arxiv.org/pdf/2305.14763), and [Kim et al., 2023](https://arxiv.org/pdf/2310.15421)). The core challenge for a paper like this is to establish construct validity, especially in light of a literature indicating that such measures are very hard to make robust and meaningful. This paper does not engage with that challenge. So I am left wondering if it is just another potentially-meaningless probing experiment.
> >
> > The other important point for validity of the experiment is on the issue of the effect of steering on oracle beliefs. You state:
> > > The dataset that we use does not evaluate oracle settings. Given that the task focuses on theory of mind – which implies taking someone else's perspective – oracle settings are less interesting as the task boils down to text comprehension, which current models can already perform well.
> >
> > This doesn't address the issue I was raising. My concern is that activation steering which improves the measured ToM result may not be an enhancement to a model's reasoning if it degrades the model's ability to do other tasks. It's kind of like if you trained a probe on the model's representations to predict the ToM outcome and then ensembled it with the model's original prediction. Of course this will improve the model's performance: you're incorporating a bunch of task-specific statistical signal. But if you do this intervention when querying the model for any other task, it would fall apart, since half of the ensemble is always just trying to do the ToM prediction task you trained it for. So the steering results would be much more convincing if you could show that they did not degrade performance on control tasks like predicting the oracle belief.
> >
> > The rest of the issues are comparatively not important for my assessment of the paper so I'll leave it at this. As of now I do not see a reason to change my score.

---

> ### Author Response · Authors · 2024-11-20
>
> > The point about steering vectors avoiding the need for training dedicated probes doesn't make sense to me. If you need a training dataset of positive and negative examples to compute a steering vector, you're in the exact same situation as someone training a probe.
>
> The important difference is that ITI needs to train one probe for every single attention head in the model, while CAA needs only one vector per layer. For example, for Llama 2 70B, ITI needs to *train* 64 * 80 = 5120 probes while CAA needs only to *compute* 80 vectors. As a nice addition to this, CAA yields improved results and strong generalisation across all tasks.
>
> > Is there a reason you used Llama 2 instead of Llama 3? 3 has been out for a while now.
>
> We used Llama 2 as it was easier to access when we performed the experiments. We are happy to add results for Llama 3 in the revision of the manuscript or in the camera-ready.
>
> > I'm confused by RQ4, about whether the probes are memorizing their training data. Does it matter? You evaluate it on held-out test data, right?
>
> Of course we are using a held-out test set. Dimensionality reduction of linear probes is used to avoid a situation where the probe might learn to rely on irrelevant patterns in the data instead of capturing meaningful relationships [1, 3]. In other words, “simply overfitting on the features
> because there are too many features” [1]. This is not an approach meant to substitute the use of a test set, but to complement it.
> \
> \
> We kindly ask if the reviewer could consider whether our clarifications support an increase in their score.
> \
> \
> References \
> [1] Alain, Guillaume. "Understanding intermediate layers using linear classifier probes." ICLR 2017. \
> [2] Gurnee, Wes, et al. "Language Models Represent Space and Time." ICLR 2024. \
> [3] Zhang, Chiyuan, et al. "Understanding deep learning (still) requires rethinking generalization." Communications of the ACM 64.3 (2021): 107-115.
> [4] Li, Kenneth, et al. "Inference-time intervention: Eliciting truthful answers from a language model." NeurIPS 2024.

---

### Note · Authors · 2024-12-16

I have read and agree with the venue's withdrawal policy on behalf of myself and my co-authors.